# Metabolic Fingerprints of Effective Fluoxetine Treatment in the Prefrontal Cortex of Chronically Socially Isolated Rats: Marker Candidates and Predictive Metabolites

**DOI:** 10.3390/ijms241310957

**Published:** 2023-06-30

**Authors:** Dragana Filipović, Julica Inderhees, Alexandra Korda, Predrag Tadić, Markus Schwaninger, Dragoš Inta, Stefan Borgwardt

**Affiliations:** 1Department of Molecular Biology and Endocrinology, “VINČA” Institute of Nuclear Sciences—National Institute of the Republic of Serbia, University of Belgrade, 11000 Belgrade, Serbia; 2Institute for Experimental and Clinical Pharmacology and Toxicology, Center of Brain, Behavior and Metabolism, University of Lübeck, 23562 Lübeck, Germany; julica.inderhees@uni-luebeck.de (J.I.); markus.schwaninger@uni-luebeck.de (M.S.); 3German Centre for Cardiovascular Research (DZHK), Partner Site Hamburg-Kiel-Lübeck, 20251 Hamburg, Germany; 4Center of Brain Behavior and Metabolism, University of Lübeck, 23562 Lübeck, Germany; 5Department of Psychiatry and Psychotherapy, Center of Brain Behavior and Metabolism, University of Lübeck, 23562 Lübeck, Germany; alexandra.korda@uni-luebeck.de; 6School of Electrical Engineering, University of Belgrade, 11000 Belgrade, Serbia; ptadic@etf.bg.ac.rs; 7Department for Community Health, Faculty of Natural Sciences, Medicine, University of Fribourg, 1700 Fribourg, Switzerland; dragos.inta@unifr.ch (D.I.); stefan.borgwardt@uksh.de (S.B.); 8Department of Biomedicine, University of Basel, 4001 Basel, Switzerland

**Keywords:** depressive-like behavior, prefrontal cortex, fluoxetine, metabolomics, ROC curve, support vector machine-linear kernel

## Abstract

The increasing prevalence of depression requires more effective therapy and the understanding of antidepressants’ mode of action. We carried out untargeted metabolomics of the prefrontal cortex of rats exposed to chronic social isolation (CSIS), a rat model of depression, and/or fluoxetine treatment using liquid chromatography–high resolution mass spectrometry. The behavioral phenotype was assessed by the forced swim test. To analyze the metabolomics data, we employed univariate and multivariate analysis and biomarker capacity assessment using the receiver operating characteristic (ROC) curve. We also identified the most predictive biomarkers using a support vector machine with linear kernel (SVM-LK). Upregulated myo-inositol following CSIS may represent a potential marker of depressive phenotype. Effective fluoxetine treatment reversed depressive-like behavior and increased sedoheptulose 7-phosphate, hypotaurine, and acetyl-L-carnitine contents, which were identified as marker candidates for fluoxetine efficacy. ROC analysis revealed 4 significant marker candidates for CSIS group discrimination, and 10 for fluoxetine efficacy. SVM-LK with accuracies of 61.50% or 93.30% identified a panel of 7 or 25 predictive metabolites for depressive-like behavior or fluoxetine effectiveness, respectively. Overall, metabolic fingerprints combined with the ROC curve and SVM-LK may represent a new approach to identifying marker candidates or predictive metabolites for ongoing disease or disease risk and treatment outcome.

## 1. Introduction

The leading cause of disability and one of the main sources of global burden of disease worldwide in adults is major depressive disorder (MDD), known as depression, a common mental disease with a complex neurobiological basis. Different hypotheses are used to understand the pathogenesis of depression, including changes in the synthesis and metabolism of monoamine neurotransmitters, mainly serotonin (5-hydroxytryptamine, or 5-HT), disturbances in their receptor function, or changes in signal transduction pathways at the post-receptor level [1,2,3]. To investigate the molecular mechanism underlying depression and evaluate the efficacy of antidepressants, we used a well-validated animal model for chronic psychosocial stress, chronic social isolation (CSIS). The depressive-like behavior induced by CSIS in rats is accepted to resemble the effects of perceived isolation in humans that may contribute to the development of depression [4,5,6]. The prefrontal cortex (PFC) is one area of the brain that experiences structural and functional changes in depression [7,8,9,10]. Its dysfunction has been associated with cognitive impairment, including decision-making and working memory, as a common symptom of depression [11]. A previous study has shown that CSIS in adult male Wistar rats compromises the hypothalamic–pituitary–adrenal (HPA) axis activity and impairs the glucocorticoid negative feedback response in the extrinsic HPA axis structure such as PFC [12,13], as found in depressive patients [14,15]. Additionally, alterations in mitochondrial dynamics, oxidative phosphorylation, mitochondrial biogenesis, and the production of antioxidant enzymes have been revealed in the PFC of serotonin transporter knockout rats [16]. Recently, a proteomics study in the PFC of CSIS rats has shown that depression-like behavior is associated with compromised mitochondrial membrane integrity; CSIS affected mitochondrial transport and energy processes as well as synaptic neurotransmission and oxidative stress [17].

Depression is frequently treated with selective serotonin reuptake inhibitors, such as fluoxetine (Flx), which improve serotonergic neurotransmission by inhibiting its reuptake transporter [18]. This drug reversed depressive-like behavior in the rat CSIS model of depression [17,19]. Effective Flx treatment of CSIS rats altered mitochondrial bioenergetics, vesicle-mediated transport, and synaptic signaling in the PFC [17,20]. Additionally, Flx stimulates neurogenesis and neural plasticity in various brain regions [21,22,23]. Although Flx has been proven to be effective in depressive patients, the response rate is low (60–70% of patients), and the mechanisms of action have not been defined yet. Moreover, the identification of metabolic biomarkers and the establishment of a strategy for their screening and application are needed. Therefore, new approaches are required for a more detailed examination of the pathobiology of depression and the mode of antidepressant action, as well as the identification of biomarker candidates for depression pharmacotherapy.

It has been shown that mental and behavioral changes have been associated with metabolic alterations [24]. Metabolomics is a study for identifying metabolite changes in cellular processes and presents characteristic small-molecule fingerprints related to the pathophysiology of depression in both clinical research [25,26] and animal experiments [27,28,29], as well as following antidepressant treatments [30,31]. Indeed, metabolic changes in the PFC of rats that showed depressive-like behavior following chronic unpredictable mild stress were found in amino acid metabolism, energy metabolism, lipid metabolism, oxidative stress, and the synthesis of neurotransmitters [29]. Hence, we investigated the application of liquid chromatography–high resolution mass spectrometry (LC–HRMS)-based untargeted metabolomics in CSIS (6-week) rats, and CSIS rats with chronic Flx treatment (lasting three weeks of 6-week CSIS), and controls. To analyze metabolomics data, univariate (*t*-test) and multivariate partial least square-discriminant analysis (PLS-DA) were performed. A classical receiver operating characteristic (ROC) curve analysis was used to assess the molecular marker performances of each metabolite for a binary outcome. For deeper data analysis, a support vector machine with linear kernel (SVM-LK), as a machine-learning classification model, was applied to identify a subset of predictive metabolites that have the potential to enable the more accurate diagnosis of a depressive phenotype or effective antidepressant treatment [32,33]. The correlation between metabolites and immobility time in the forced swim test (FST) was investigated to test whether the level of metabolites could reflect behavior despair in the CSIS model. To date, no studies have examined the PFC metabolic fingerprints of adult male CSIS rats in combination with Flx treatment. In addition, this is the first study where obtained metabolic fingerprints were used for the identification of marker candidates for the designation of depressive-like behavior following CSIS and effective Flx treatment in CSIS rats, and the most-contributing predictive metabolites for binary-group classification (CSIS vs. Control and CSIS + Flx vs. CSIS).

## 2. Results

### 2.1. Behavioral Testing 

The FST results are presented in Figure 1. For immobility time, a significant main effect of CSIS (F_1.23_ = 13.07, *p* ˂ 0.01) and effects of time (F_2.46_ = 15.10, *p* ˂ 0.001), CSIS × time (F_2.46_ = 4.34, *p* ˂ 0.05), and Flx × time (F_2.46_ = 8.43, *p* < 0.001) were found. A significant increase in immobility time in CSIS + Flx and CSIS at the 3-week test compared to baseline (*** *p* < 0.001) was revealed. At the 6-week test, only the CSIS group differed from baseline (*** *p* < 0.001). 

For swimming behavior, a significant main effect of CSIS (F_1.23_ = 18.42, *p* ˂ 0.001) and effects of time (F_2.46_ = 7.56, *p* ˂ 0.01), CSIS × time (F_2.46_ = 4.53, *p* ˂ 0.05), and Flx × time (F_2.46_ = 9.88, *p* < 0.001) were observed. A significant decrease in swimming time in CSIS + Flx and CSIS at the 3-week test compared to baseline (* *p* < 0.05) was found. Only the CSIS remained significantly decreased from baseline at the 6-week test (*** *p* < 0.001). No significant main effects of CSIS or Flx treatment on climbing behavior were observed. The reduced climbing of Control + Flx and CSIS rats at the end of the 6th week compared to the baseline (* *p* < 0.05) was found.

### 2.2. PFC Metabolic Fingerprints following CSIS and/or Effective Flx Treatment and Controls

In the LC–HRMS analysis, a total of 117 metabolites in each sample were identified (Appendix A). The list of statistically significant metabolite changes is presented in Table 1.

### 2.3. Multivariate Data Analysis

PLS-DA was performed to determine the discrimination between groups based on the metabolomics dataset. Dimensionality reduction resulted in the organization of samples based on two components. The key parameters, R^2^ and Q^2^, in pair-wise groups were higher than 0.5 (Table 2), indicating that models were robust and had good fitness and prediction. The Control + Flx was clearly distinguished from the Control group in the PLS-DA plot (Figure 2a). Additionally, the PLS-DA score plot (Figure 2b) shows that the CSIS group had a distinctive metabolic profile from the control group, and a clear separation was also observed between the CSIS + Flx and CSIS group (Figure 2c). 

Moreover, Appendix A provide dendrogram (A) and heatmaps (B) of the hierarchical cluster analysis, i.e., metabolite changes for pair-wise comparisons

### 2.4. Marker Candidate Identification

ROC curve based on the area under the curve (AUC) values revealed metabolites with the best marker preferences in the rat PFC of depressive-like behavior and effectively Flx-treated rats (Table 3).

According to ROC analysis, myo-inositol with AUC = 1 had the best molecular candidate preferences for CSIS group designation (Figure 3a). Sedoheptulose-7-phosphate (1), hypotaurine (0.98214) and acetyl-L-carnitine (0.96429) had the greatest AUC (in parenthesis) values of classical ROC analysis and were the most significant marker candidates following effective Flx treatment in CSIS rats (Figure 3b–d). 

### 2.5. SVM Classification

The SVM-LK classifier showed the optimal classification performances with accuracy of 61.50% in predicting depressive-like behavior in CSIS rats with a panel of seven contributing PFC metabolites. An accuracy of 93.3% with a panel of 25 contributing PFC metabolites was achieved to classify Flx-treated CSIS rats from CSIS rats, representing normalized behavior and Flx effectiveness. The SVM-LK classifiers with the most contributing metabolites are presented in Table 4. 

### 2.6. Correlation of Behavioral Phenotype with the Metabolomics Data

A Pearson correlation analysis was conducted between all metabolites and immobility time at 6 weeks in FST. The results are shown in Table 5.

A moderate positive correlation of myo-inositol with immobility time in the FST between CSIS and control groups, and a moderate negative correlation of sedoheptulose-7-phosphate with immobility time in the FST between CSIS + Flx vs. CSIS, were revealed. These findings show that altered metabolite levels could reflect depression-like behavior in CSIS rats.

## 3. Discussion

The obtained metabolic fingerprints revealed metabolites that were significantly altered in depressive-like behavior following CSIS and effective Flx treatment. We also proposed marker candidates and a panel of predictive metabolites, contributing the most to group designation or binary classification, respectively.

Following CSIS stress, rats showed depressive-like behavior, assessed by an increased immobility time in FST, as a measure of behavior despair. This result is consistent with previous studies of CSIS-induced depression-like behavior assessed by sucrose preference and open field tests [17,34,35]. Chronic Flx treatment (15 mg/kg/day) significantly reduced the immobility behavior of CSIS rats, implying that Flx reversed depressive-like symptoms in stressed rats, indicating an antidepressant effect.

A possibly affected pathway related to depressive-like behavior might be the phosphoinositide pathway, with myo-inositol (MI) being significantly altered between CSIS and control. It is a component of phosphatidylinositols, membrane phospholipids that can also participate as second messengers [36]. In the brain, MI is synthesized de novo from D-glucose-6-phosphate by endothelial cells located at the blood–brain barrier or through interconversion between inositol derivatives. It can be transferred across the blood–brain barrier, originating from the diet or other organs [37,38]. As it is more prevalent in astrocytes, a major glial cell type, than in neurons [39], the elevation of MI level in our study might reflect glial activation, indicating changes in astroglia cell metabolism, which have the ability to release pro-inflammatory cytokines and free radicals, causing functional brain impairment [40,41,42,43]. In keeping with previous studies, CSIS resulted in the impairment of rat PFC function by pro-inflammatory mediators such as interleukin-1 beta and tumor necrosis factor alpha or compromised glutathione antioxidant defense [17,44]. The revealed significant, positive correlation between increased immobility behavior and MI level (r = 0.5932) is compatible with the idea that MI is at least in part involved in the depressive-like behavior of CSIS rats. Additionally, the increased level of MI might change second messenger systems, which could lead to alterations in inositol trisphosphate signaling and its role in intracellular calcium mobilization [45], which can trigger apoptosis [46,47], already confirmed in the PFC of CSIS rats [7,48]. 

Although differential MI content has been reported depending on the model species and brain regions [49,50,51,52] or age [53,54,55], MI has been suggested as a metabolic marker for depression [56]. In line with this, the classical ROC analysis in our study identified MI as the best candidate to discriminate CSIS from controls. In terms of showing predictive metabolites for CSIS classification from controls, SVM-LK revealed a panel of seven metabolites, of which four were amino acids such as tyrosine, asparagine, valine, and serine that may contribute to or reflect depression-like behavior. The strongest decrease was found in valine level (FC 0.84). Given that valine is associated with cognitive performance [57], a decrease in its content is likely connected to changes in cognitive function as a hallmark of depression. Conversely, the highest increase was revealed in succinate levels (FC 1.26) contributing the most to group designation. Succinate is a substrate of the TCA cycle and higher levels may indicate the stimulation of the TCA cycle to increase energy demand and to aid cells in coping with stress.

Effective Flx treatment in CSIS rats caused more obvious metabolic perturbations in the PFC, whereby six metabolites were found different by univariate analysis. One of the most prominent changes was the significant elevation of acetyl-L-carnitine (ALC) which also had one of the best molecular candidate preferences for Flx efficacy group designation (AUC 0.964). This metabolite has been demonstrated to exert antidepressant effects by improving mitochondrial energy, the regulation of neurotransmission, and neural plasticity [58]. Thus, it facilitates the transport of activated long-chain fatty acids into the mitochondria to undergo subsequent β-oxidation. Generated acetyl-CoA increases mitochondrial energy production by entering the TCA cycle, or it can be incorporated into glutamate, glutamine, and GABA [58,59]. The capability of ALC treatment to decrease oxidative stress has also been reported [60]. Given that mitochondrial oxidative stress has been described in the PFC of CSIS rats [17], we may suppose that an increase in ALC levels aimed to restore normal mitochondrial function and rats’ behavioral normalization. Preclinical and clinical data indicate that ALC is more rapidly effective than Flx and promotes structural plasticity in the limbic brain region [61,62,63]. Furthermore, it has been demonstrated that ALC and Flx have equal antidepressant efficacy, with ALC efficacy being noticed after 3 days of the treatment and lasting for two weeks after cessation of its application [64,65]. Additional studies using CSIS paradigms would provide more insight into the possible therapeutic effectiveness of ALC in protecting the brain.

The content of sedoheptulose-7-phosphate, an intermediate in the pentose phosphate pathway, was increased following effective Flx treatment in CSIS as well as control rats. Its increased levels may be linked to the production of ribose 5-phosphate, which is needed for nucleotide synthesis, or NADPH, which provides electrons to antioxidants combating harmful oxygen molecules [66]. In addition, elevated contents of stachydrine and 2-hydroxyglutaric acid, which are implicated in the metabolism of amino acids and energy production, respectively, were found in Flx-treated controls. Although Flx did not alter the behavior phenotype in control rats, altered metabolites may suggest adaptive cellular responses to chronic Flx treatment. Additionally, hypotaurine, as the metabolic precursor of taurine, whose content was increased in effectively Flx-treated rats, may also act as an antioxidant by scavenging highly reactive hydroxyl radicals [67]. Based on the AUC of classical ROC analysis, sedoheptulose-7-phosphate (1), hypotaurine (0.982), and ALC (0.964) were suggested as marker candidates for differentiating effective Flx treatment behavioral outcomes. SVM-LK achieved a better performance in predicting effective Flx treatment in CSIS rats compared to CSIS than predicting CSIS compared to control. The best accuracy of classification was attained for a panel of 25 metabolites, including decanoylcarnitine and L-carnitine, both mitochondrial metabolites. Although predictive metabolites were not significantly changed by univariate analysis, the pattern of recognition of a particular class is influenced by absolute values of the most significant, or predictive, variables as well as the existence of complex interactions between these variables [68]. Altogether, this approach would require a substantially larger number of biological replicates and training-set size for a more accurate distinction between the examined groups.

Furthermore, we found a decline in the contents of xanthosine, riboflavin, and hexanoylcarnitine following effective Flx treatment in CSIS rats (Table 1). Given that xanthosine may be a result of higher oxidative stress caused by purine catabolism [69], we can assume that Flx may be associated with a reduction in oxidative stress and consequently a decline in xanthosine levels. The same tendency was observed for riboflavin, also known as vitamin B2, that neurons and astrocytes obtain from the blood through the blood–brain barrier. According to the literature, Flx suppressed the metabolism of riboflavin due to its involvement as an important cofactor in tryptophan metabolism and a crucial methyl donor in the conversion of homocysteine [70]. Given that we did not detect changes in the tryptophan or cysteine metabolic pathways in our study, a reduced riboflavin level may arise from its decreased uptake from the bloodstream. Moreover, a decrease in hexanoylcarnitine levels was detected, and it was also revealed as one of the marker candidates by the classical ROC/AUC value (0.964) to discriminate effective Flx treatment in CSIS rats. This metabolite promotes the transport of medium-chain fatty acids into the mitochondria. Moreover, it may be concluded that alterations of aforementioned metabolites reversed the behavioral alterations following CSIS and were involved in the effective Flx treatment in CSIS rats. The limitation of the present study is that only six or eight animals per group were investigated. Increased animal numbers should be considered for future experiments.

## 4. Material and Methods

### 4.1. Animals

We used adult male Wistar rats (2.5 months of age, 300–350 g weight) bred in the Animal Facility of “VINČA” Institute of Nuclear Sciences, National Institute of the Republic of Serbia, University of Belgrade. Rats were kept under standard conditions in groups of up to four per cage with a 12 h light/dark cycle, a humidity level of 55 ± 10%, a temperature of 20 ± 2 °C, and free access to food (commercial rat pellets) and water ad libitum. All experimental procedures were approved by the Ethical Committee for the Use of Laboratory Animals of the “VINČA” Institute of Nuclear Sciences, National Institute of the Republic of Serbia, University of Belgrade, which follows the guidelines of the EU-registered Serbian Laboratory Animal Science Association (SLASA). The study protocol was approved by the Ministry of Agriculture, Forestry, and Water Management—Veterinary Directorate, ethics committee, license number 323-07-02256/2019-05. Rats were monitored daily.

### 4.2. Fluoxetine Hydrochloride Administration

Flunisan tablets (containing 20 mg of fluoxetine hydrochloride, Hemofarm, Vršac, Serbia) were crushed, dissolved in distilled, sterile water with the aid of ultrasound, and filtered through Whatman No. 42 filter paper. Ultra-Performance Liquid Chromatography analysis was used to determine the concentration of Flx solution [71]. We recorded a total loss of 25% in drug concentration throughout the preparation method, which was accounted for during drug administration (15 mg/kg/day) [35]. The solution of Flx was administered according to rat weight measured once a week. Flx serum concentrations were similar to those reported in the serum of patients effectively treated with Prozac [35,72]. 

### 4.3. Experimental Design

The experimental design is graphically represented in Figure 4. A CSIS model was employed as previously described [17]. At the onset of the experiment (week 0), rats (*n* = 50) were randomly assigned into two groups: control (*n* = 20, housed in groups of up to four) and CSIS (*n* = 30, housed individually, with no tactile or visual contact). For the first 3 weeks, rats were not exposed to any additional experimental procedures. During the second 3-week period, half of each group of rats was treated daily with intraperitoneal (i.p.) Flx solution (15 mg/kg/day) (Control + Flx and CSIS + Flx); the remaining rats were administered daily i.p. injections of physiological solution (Control + Vehicle and CSIS + Vehicle). The assessment of depressive-like behavior and effectiveness of Flx treatment in rats were performed according to the results of immobility time in the FST. The test was performed before the start of the experiment (week 0, baseline) and at the end of the 3rd and 6th weeks. Given that depression in humans induced by social factors is associated with a higher risk of mortality in males [73] and that metabolic changes depend on the estrous cycle [74], the experiments were conducted only in male rats.

### 4.4. Forced Swim Test

The FST was performed to evaluate the depressive-like behavior of rats undergoing the CSIS procedure and the antidepressant-like effect of Flx, as previously described [17]. Rats were individually placed into plexiglass cylinders (height 45 cm, diameter 28 cm) filled with water (24 ± 1 °C) up to a height of 33 cm. During the 5 min long test, immobility, climbing, and swimming were recorded [75], and results were analyzed by two observers blinded to the experimental conditions. Immobility was defined as floating in the water without making any effort but making movements to keep one’s head above the surface. Rats following CSIS that showed immobility increases >20% at the end of the 3rd and 6th week of testing compared to baseline, were designated as CSIS. Flx-treated CSIS rats that showed a decrease in immobility behavior >20% at the end of the 6th week relative to CSIS at the end of the 3rd or 6th week, were designated as responsive to Flx treatment. Rats following CSIS which showed no immobility increase compared to baseline (CSIS resilient), and CSIS + Flx rats which showed no immobility decline at the end of the 6th week (Flx resilient) compared to CSIS rats, were not included in the current study. The final number of animals per group was 6–8.

### 4.5. Metabolomics Analysis by LC–HRMS 

#### 4.5.1. Optimization of Sample Preparation for LC–HRMS Analysis

Once all the behavioral testing was completed, the rats were anesthetized with a mixture of ketamine/xylazine (120/16 mg/kg) and sacrificed by decapitation. The PFC was dissected from the brain on ice, frozen with liquid nitrogen rapidly, and stored at −80 °C until further analysis.

The frozen PFC samples were pulverized using the Cellcrusher (Kisker Biotech GmbH & Co. KG, Steinfurt, Germany). The tissue was weighed (~10 mg) and metabolic profiling was performed as described previously [76] with minor modifications. Briefly, metabolites were extracted using 500 µL of methanol/acetone/acetonitrile/water (1/1/1/0.75, *v*/*v*/*v*/*v*) containing 2.5 µM Metabolomics Amino Acid Mix Standard (Cambridge Isotope Laboratories, Andover, MA, United States). After mixing for 15 min at 4 °C at 1000 rpm (ThermoMixer Eppendorf), samples were sonicated for 1 min and vortexed for 10 s. After incubation for 2 h at −20 °C, samples were centrifuged for 10 min at 14,000 rpm at 4 °C. The collected supernatants were evaporated to dryness in a vacuum concentrator (SpeedVac Concentrator, ThermoFisher Scientific, Waltham, MA, USA). The dry extracts were reconstituted in 50 µL of methanol/acetonitrile (1:1) and vortexed for 15 sec followed by centrifugation at 14,000 rpm for 10 min at 4 °C. The supernatants were transferred to LC/MS vials, and LC–HRMS analysis was performed. Pooled quality control samples (QC) were prepared in the same manner to ensure data quality and linearity. All solvents were of LC-MS-grade quality and were purchased from Merck (Darmstadt, Germany).

#### 4.5.2. Metabolic Profiles Analyzed by LC–HRMS 

LC–HRMS analysis was performed using a Dionex Ultimate 3000 RS LC-system coupled to an Orbitrap mass spectrometer (QExactive, ThermoFisher Scientific, Bremen, Germany) equipped with a heated-electrospray ionization (HESI-II) probe. Extracted metabolites were separated on a SeQuant ZIC-HILIC column (150 × 2.1 mm, 5 µm) using water with 5 mM ammonium acetate as eluent A and acetonitrile/eluent A (95:5, *v*/*v*) as eluent B. The gradient elution was set as follows: isocratic step of 100% B for 3 min, 100% B to 60% B in 15 min, held for 5 min, returned to initial conditions in 5 min and held for 5 min. The flow rate was 0.5 mL/min. Data was acquired based on a full MS/data-dependent MS^2^ (top 10) experiment. Data processing was performed using Compound Discoverer 3.1 (ThermoFisher, CA, USA). Metabolites were identified based on exact mass, retention time, fragmentation spectra and isotopic pattern. We used an in-house library [76] as well as the online library mzCloud. The final output data includes the compound name, retention time (RT), exact mass-to-charge (*m*/*z*) ratio, and peak area.

#### 4.5.3. Metabolite Data Statistic and Analysis

We used the web-based tool MetaboAnalyst5 (http://www.metaboanalyst.ca/) (accessed on 17 March 2023) to perform statistical analysis of metabolome data. Briefly, peak areas were normalized by the total sum scaling method followed by a log transformation (base10). Metabolites were further applied to the univariate analysis for pair-wise group comparisons using a *t*-test and FC. Metabolites with FDR-adjusted *p*-values of <0.05 and FC thresholds of >1.5 were considered statistically significant [77]. Then, multivariate analysis was performed using PLS-DA, which maximizes the discrimination between the two groups by incorporating known classification information. Estimated values of R^2^ were used to explain the model fitness, and Q^2^ was described for the predictive accuracy of its class mode. 

### 4.6. Identification of Marker Candidates 

For assessing the molecular marker performance for each metabolite as a marker candidate, firstly ROC curve analysis AUC evaluation were applied using MetaboAnalyst 5.0 Biomarker Analysis tool [78]. Metabolites with AUC > 0.9 were discussed in terms of marker capacity. The proportion of correctly classified rats with effective Flx treatment is measured by sensitivity, or the true-positive rate; the proportion of correctly identified control subjects is measured by specificity, or the true-negative rate.

### 4.7. SVM-LK-Based Binary Classification

SVM is one of the most prominent supervised machine learning algorithms, which shows the best predictive performance (balanced, accuracy, sensitivity, and specificity for each pairwise combination of variables, compared to other machine learning approaches in diseases and drug treatment [79,80,81,82] and also in precision medicine [32]. Accuracy is defined as the percentage of correctly classified samples. Therefore, to obtain the best possible diagnostic model, and to account for possible interactions between the features (which are ignored by the ROC/AUC analysis), we selected the input features for the SVM model using a greedy forward-selection method. This method selects feature combinations that maximize the predictive accuracy of the model in the CV1 test data, stopping at 50% of features. Moreover, a 10-times-repeated 3 × 3 nested cross-validation procedure was conducted with SVM-LK in order to avoid information leakage between subjects used for training and validating the models, and to enhance the generalizability of the models in new data by eliminating biased estimation. Further details are given in Appendix A.

### 4.8. Statistical Analysis

The behavioral data were analyzed with a three-way repeated-measures ANOVA, factor treatment (levels: vehicle and Flx), conditions (levels: control and CSIS) and test as a repeated measure (levels: baseline (weeks 0, 3, and 6) using Statistica 12. Significant differences between the groups were examined using Duncan’s post hoc test. The number of individual measurements was *n* = 6–8. In order to explore whether the intensity of metabolites could reflect despair in the CSIS model, the Pearson correlation analysis was used to identify the metabolites and immobility time at the 6th week in FST.

## 5. Conclusions

In summary, distinct PFC metabolic fingerprints of CSIS rats and/or following effective Flx treatment were revealed by LC–HRMS. An increased content of MI following CSIS may indicate depressive-like behavior. MI, which is involved in the phosphoinositide pathway, was also selected as a marker candidate for CSIS. Sedoheptulose-7-phosphate, hypotaurine, and ALC may be marker candidates for the treatment effect of Flx. A panel of 7 or 25 predictive metabolites, obtained by SVM-LK, could be used for binary group classification (CSIS vs. Control and CSIS + Flx vs. CSIS, respectively). Moreover, identified rat PFC-metabolite marker candidates, along with predictive metabolites and possibly involved pathways, may further elucidate the molecular mechanisms of a depressive phenotype and a mode of Flx action. 

## Figures and Tables

**Figure 1 ijms-24-10957-f001:**
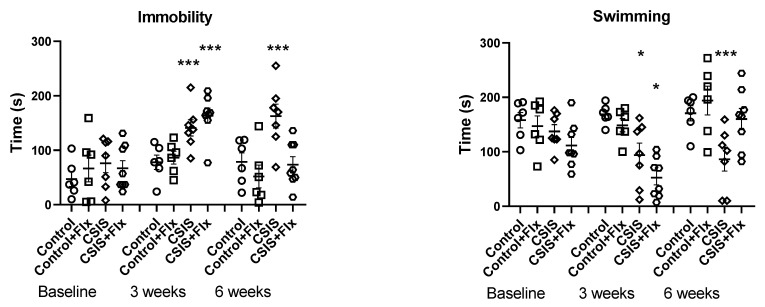
Behavioral results in control, fluoxetine-treated controls (Control + Flx), chronic social isolation (CSIS) and fluoxetine-treated CSIS (CSIS + Flx) rats in the forced swim test (FST) at baseline and at the end of the 3rd and 6th week. Differences between different groups compared to controls (baseline) were considered statistically significant at *** *p* < 0.001, * *p* < 0.05. Immobility—CSIS or CSIS + Flx (3 weeks) vs. CSIS (baseline) as well as CSIS (6 weeks) vs. CSIS (baseline) *** *p* < 0.001; Swimming—CSIS or CSIS + Flx (3 weeks) vs. CSIS (baseline) * *p* < 0.05 and CSIS (6 week) vs. CSIS (baseline) *** *p* < 0.001; Climbing—Control + Flx and CSIS (6 weeks) vs. Control + Flx and CSIS (baseline) * *p* < 0.05. Significant differences between groups obtained via a three-way repeated-measures ANOVA, followed by Duncan’s post hoc test. Data are expressed as the mean ± standard deviation (± SDEV); n = 6–8 rats per each group.

**Figure 2 ijms-24-10957-f002:**
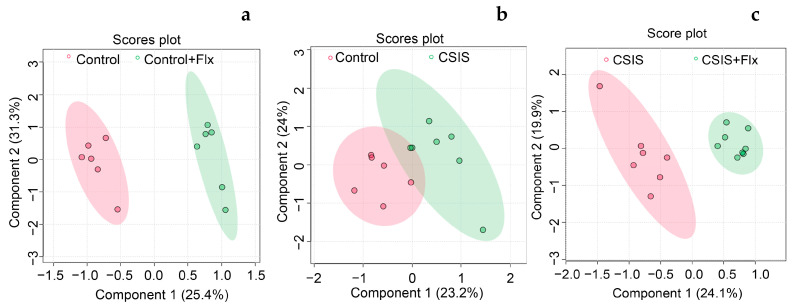
PLS-DA score plots show separation based on specific metabolic profiles of Control + Flx vs. Control (**a**), CSIS vs. Control (**b**), and CSIS + Flx vs. CSIS (**c**). Each dot represents the function of the metabolic profile of an individual sample.

**Figure 3 ijms-24-10957-f003:**
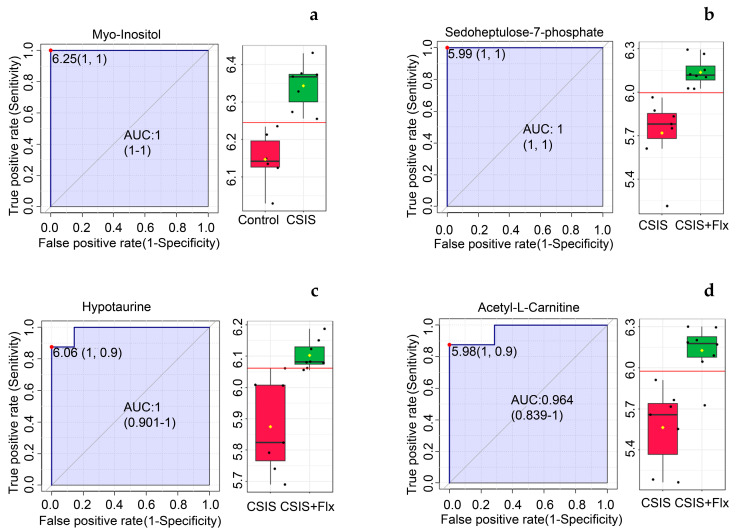
PFC molecular marker candidates in depressive-like behavior following CSIS (**a**) and effectively Flx-treated animals (**b**–**d**) based on a classical ROC curve with AUC values. ROC curves are presented with 95% confidence interval and AUC values. Box-and-whisker plots display individual variable distributions within each group. Red dots (ROC curves) and red lines (box-and-whisker plots) represent the optimal cut-off value between the groups.

**Figure 4 ijms-24-10957-f004:**
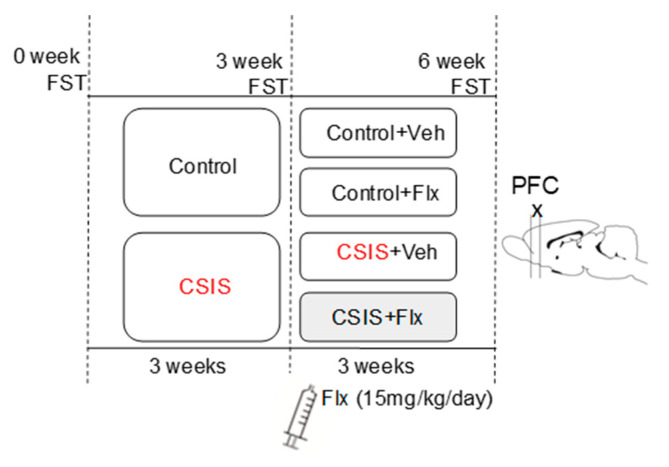
Experimental design of the study. CSIS—Chronic Social Isolation; Flx—fluoxetine; FST—forced swim test; PFC—prefrontal cortex.

**Table 1 ijms-24-10957-t001:** List of significantly changed metabolites following Flx treatment in control, CSIS and effective Flx treatment in CSIS groups in the rat prefrontal cortex detected by LC-HRMS. False discovery rate (FDR)-adjusted *p*-values (*t*-test) of <0.05 and fold change (FC) thresholds of >1.5.

			Control + Flx vs. Control			CSIS vs. Control			CSIS + Flx vs. CSIS		
Retention Time (min)	Metabolites	FC	*p*-Value	*p*-Adjusted	FC	*p*-Value	*p*-Adjusted	FC	*p*-Value	*p*-Adjusted	Metabolic Pathway
11.28	N-acetyl-L-arginine	0.27	1.00 × 10^−5^	1.20 × 10^−3^							Amino acid metabolism
5.57	Xanthine	0.61	1.95 × 10^−3^	3.80 × 10^−2^							Purine metabolism
9.50	N1-methyl- nicotinamide	0.65	7.26 × 10^−4^	2.12 × 10^−2^							Energy metabolism
14.25	Sedoheptulose-7-phosphate	2.24	1.09 × 10^−4^	2.56 × 10^−2^				2.4	6.89 × 10^−4^	2.02 × 10^−2^	Energy metabolism
4.16	2-hydroxy glutaric acid	2.28	6.84 × 10^−4^	2.12 × 10^−2^							Energy metabolism
4.62	Indoxylsulfate	2.57	2.49 × 10^−3^	4.16 × 10^−2^							Organic acids and derivatives
10.09	Stachydrine (proline betaine)	5.08	3.10 × 10^−5^	1.80 × 10^−3^							Amino acid metabolism
11.98	Myo-inositol				1.56	2.69 × 10^−4^	3.15 × 10^−2^				Inositol phosphate metabolism
7.10	Hexanoyl carnitine							0.45	2.60 × 10^−4^	2.02 × 10^−2^	Lipid metabolism
7.98	Xanthosine							0.62	6.83 × 10^−4^	2.02 × 10^−2^	Purine metabolism
5.32	Riboflavin							0.64	9.14 × 10^−4^	2.14 × 10^−2^	Riboflavin metabolism
11.65	Hypotaurine							1.62	1.16 × 10^−3^	2.25 × 10^−2^	Lipid metabolism
8.75	Acetyl-L- carnitine							3.31	5.24 × 10^−4^	2.02 × 10^−2^	Lipid metabolism

**Table 2 ijms-24-10957-t002:** PLS-DA classifier performances.

Group Comparison	No of Component	R^2 a^	Q^2 b^	Accuracy
Control + Flx vs. Control	5	0.99953	0.92444	1
CSIS vs. Control	5	0.99587	0.29245	0.84615
CSIS + Flx vs. CSIS	5	0.99636	0.86544	1

^a^ Measure of goodness of fit of the model; ^b^ Measure of predictive ability of the model.

**Table 3 ijms-24-10957-t003:** List of PFC metabolites with the best marker preferences for depressive-like behavior and effectively Flx-treated rats.

Metabolites		CSIS vs. Control	
	AUC	*p*-Value	Fold Change
Myo-Inositol	1.000	2.69 × 10^−4^	1.56
Methylnicotinamide	0.95238	2.40 × 10^−3^	0.75
cAMP	0.92857	1.13 × 10^−2^	1.66
NAD	0.90476	2.03 × 10^−2^	1.76
Sedoheptulose-7-phosphate	1	6.89 × 10^−4^	2.40
Hypotaurine	0.982140	1.16 × 10^−3^	1.62
Riboflavin	0.982140	1.29 × 10^−3^	0.64
Acetyl-L-carnitine	0.964290	5.24 × 10^−4^	3.31
Hexanoylcarnitine	0.964290	2.60 × 10^−4^	0.45
Xanthosine	0.946430	6.83 × 10^−4^	0.62
Aconitate	0.928570	4.69 × 10^−3^	0.71
Cytosine5	0.910710	2.98 × 10^−3^	1.39
5-Methylcytosine	0.910710	5.77 × 10^−3^	0.76
Myo-Inositol	0.910710	3.57 × 10^−3^	0.76

**Table 4 ijms-24-10957-t004:** SVM-LK-based binary classification performance for pair-wise comparisons of the PFC metabolite samples.

CSIS vs. Control		CSIS + Flx vs. CSIS	
Accuracy	61.50%	Accuracy	93.30%
Sensitivity	66.70%	Sensitivity	85.70%
Specificity	57.10%	Specificity	100.0%
Balanced Accuracy	61.90%	Balanced Accuracy	92.90%
Predictive metabolites		Predictive metabolites	
Metabolites	FC	Metabolites	FC
Tyrosine	0.91	PLK	0.82
Methylnicotinamide	0.75	Phenylalanine	0.91
Hypoxanthine	0.78	Decanoylcarnitine	0.93
Asparagine	1.16	Histidine	0.90
Succinate	1.26	Pantothenic acid	0.97
Valine	0.84	Tyrosine	0.85
Serine	1.15	Inosine monophosphate	0.78
		Alanine	0.95
		Phosphatidylcholine	1.04
		Glycerophosphocholine	0.71
		Fumarate	0.87
		Thymine	0.92
		Carnitine	0.88
		Cytidinemonophosphate	1.12
		Creatine	1.04
		Cystathionine	0.75
		Adenosinediphosphoribose	0.81
		N-Acetylaspartylglutamicacid	0.73
		C6sugaralcohol	0.71
		Succinate	1.10
		Indoxylsulfate	1.65
		Cytidinediphosphocholine	1.19
		Guanosinemonophosphate	1.19
		Dihydroxyacetone phosphate	1.24
		Xanthine	0.77

**Table 5 ijms-24-10957-t005:** Statistically significant Pearson correlation (0.4 < r < −0.4), *p* < 0.05, between the PFC metabolites and immobility time in the FST.

Metabolites	*r*	*p*
Sedoheptulose-7-phosphate	−0.5698	3.70 × 10^−3^
Indoxylsulfate	−0.4942	1.41 × 10^−2^
Cytosine	−0.4642	2.24 × 10^−2^
C6H13O9P	−0.4622	2.30 × 10^−2^
Urocanic acid	−0.4517	2.70 × 10^−2^
Saccharopine	0.4093	4.71 × 10^−2^
Adenosinediphosphoribose	0.4121	4.54 × 10^−2^
Acetylcholine	0.4362	3.31× 10^−2^
Adenine	0.4585	2.43 × 10^−2^
Guanosine	0.4667	2.15 × 10^−2^
Acetylarginine	0.4669	2.15 × 10^−2^
NAD	0.5001	1.28 × 10^−2^
Riboflavin	0.5359	7.00 × 10^−3^
cAMP	0.5521	5.20 × 10^−3^
Myo-Inositol	0.5932	2.30 × 10^−3^

## Data Availability

Metabolomics data are available in MassIVE. The ID of the dataset is MSV000092101. https://massive.ucsd.edu/.

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
