# Peer review of "Metabolic Fingerprints of Effective Fluoxetine Treatment in the Prefrontal Cortex of Chronically Socially Isolated Rats: Marker Candidates and Predictive Metabolites"

_ijms, 2023, doi:10.3390/ijms241310957_

Round 1

Reviewer 1 Report

Interesting work. Even though this work is overall descriptive, I appreciated the way the experiments have been performed and the different approaches used, and it provides a snapshot of the metabolic status of pfc in the different experimental groups trying to link some alterations in metabolites levels to the depressive behavior. However, I have some comments.

Abstract

In the sentence “which were identified as potential markers”, markers of what? The sentence should be completed.

In the sentence: “We identified 4 or 10 marker candidates with ROC curve greater than 0.9 for CSIS or fluoxetine effectiveness designation” I don’t understand why the authors say 4 OR 10, the markers should be or 4 or 10, or at least the authors should define a range from 4 to 10, but still the numbers of markers is extremely general. This sentence should be rephrased because is not clear. The same comment right below, these metabolites are 7 or 25?

The sentence: “for group designation or classification” is too general. Overall, in the abstract I should be able to understand the main results of the work, together with the approach (that is clear), and with the outcome that this work is supposed to provide, thus saying “for group designation or classification” I cannot understand clearly for what group or which classification these approaches are useful in the context of depression.

Introduction

Page 2 line 57-58, is the sentence on the. HPA axis related in somehow to the prefrontal cortex? Because it is not clear the conceptual jump from the prefrontal cortex to the HPA axis. And similarly there is a gap between the HPA axis sentence and the mitochondrial membrane integrity of the pfc. This to say that the HPA axis is for sure involved in depression, but it is inappropriately placed in the introduction interrupting the sentences on the pfc.

In the section mentioning serotonin I would suggest in the introduction to cite a work on SERT and depression from Brivio et al., a group that is active in this field.  

Results:

All the bar charts should be replaced with scatter dot plots for the sake of clarity and transparence of animal numbers among the different experimental groups. 

Figure 1: it should be represented in the figure when the treatment with Flx has started. 

Discussion

Why the authors use the term “effective” when referring to fluoxetine treated rats? Are the authors saying that the treatment was not effective in all the animals? If yes, please clarify whether all the animals developed or not the depressive phenotype (resilient, vulnerable) and if anyone was resistant to the treatment. 

Line 230 there is a full stop in the middle of the sentence.

I suggest a minor English grammar revision. 

Author Response

Dear reviewer,

We thank you for your comments and suggestions, which helped us to improve the manuscript. We agree with all your comments and suggestions, and we corrected the manuscript point by point accordingly. The revised manuscript includes the changes marked up using the track changes according to the journal requirements

Reviewer 2 Report

Title:   “Metabolic Fingerprints of Effective Fluoxetine Treatment in the 2 Prefrontal Cortex of Chronically Socially Isolated Rats: Marker 3 Candidates and Predictive Metabolites “

Authors: Dragana Filipović et al.

A Reviewer comment:

The manuscript describes the untargeted metabolomics of the prefrontal cortex of rats exposed to chronic social isolation, a rat model of depression, and/or fluoxetine treatment using liquid chromatography-high resolution mass spectrometry. The manuscript contains something new and potentially deserves publication but there is a few major/minor points that I would like to see the authors address/clarify.

Major points:

1. Researchers should conduct parallel studies on females. It is generally known that depression produces different biological symptoms in men and women. https://www.nature.com/articles/s41467-021-27604-x

2. The number of animals used for behavioral studies should be at least 10 animals in each experiment.

3. The correlation between increased immobility behavior and MI level r=0.5932 is not high. 

Author Response

Dear reviewer,

We thank you for your comments and suggestions, which helped us to improve the manuscript. We agree with all your comments and suggestions, and we corrected the manuscript point by point accordingly. The revised manuscript includes the changes marked up using the track changes according to the journal requirements.
